# Associations between COVID-19 infection, symptom severity, perceived susceptibility, and long-term adherence to protective behaviors: The Los Angeles pandemic surveillance cohort study

Chun Nok Lam[1]*, Nikhilesh Kumar[1], Shirin Emma Herzig[1], Jennifer B. Unger[1], Neeraj Sood[1,2,3]

1 Keck School of Medicine, University of Southern California, Los Angeles, California, United States of America, 2 Schaeffer Center for Health Policy and Economics, University of Southern California, Los Angeles, California, United States of America, 3 Sol Price School of Public Policy, University of Southern California, Los Angeles, California, United States of America

* Chunnok.lam@med.usc.edu

## Abstract

### Background

During the COVID-19 pandemic, protective behaviors like mask wearing or social distancing were encouraged to limit viral spread. While pandemic fatigue is tied to the reduction of protective behaviors over time, little evidence exists examining predictors of long-term protective behaviors after recovering from COVID-19.

### Purpose

This study investigates the association between COVID-19 infection status and future use of protective behaviors.

### Methods

We analyzed data from 676 adults who completed questionnaires in May 2021 and January 2023 as part of the Los Angeles Pandemic Surveillance Cohort Study. Measures included self-reported COVID-19 infection status and symptom severity, and mask wearing, hand washing, social distancing and perceived susceptibility to COVID-19. We performed Wilcoxon signed-rank tests, ordinal logit regression models, and mediation analysis to assess behavior change, associations, and whether perceived susceptibility mediated the effects.

### Results

The use of protective behaviors declined significantly from baseline to follow-up. Self-reported asymptomatic or mild COVID-19 infection was associated with less social distancing (aOR=0.57, 95% CI [0.35, 0.92]), less mask wearing (aOR=0.63, 95% CI

**Data availability statement:** Data cannot be shared publicly because the survey data was collected through a proprietary database belonging to LRW, a Material Company. Please contact Jennifer Holland at jholland@isacorp. com for the contract details and process to access this data.

**Funding:** Peter G. Peterson Foundation, Conrad N. Hilton Foundation, Office of the President of University of Southern California, Los Angeles County Department of Public Health, Centers for Disease Control and Prevention, Keck School of Medicine of USC, W.M. Keck Foundation. Funders had no role in study design, data collection and analysis, decision to publish, or preparation of the manuscript.

**Competing interests:** The authors have declared that no competing interests exist.

[0.40, 0.99]), and lower perceived susceptibility (a$\beta$ = −0.17, 95% CI [−0.33, −0.02]) at follow-up. Moderate or severe COVID-19 infection was associated with less mask wearing (aOR=0.55, 95%CI [0.38, 0.81]). Perceived susceptibility to COVID-19 mediated 15% of the effect of mild COVID-19 infection on mask wearing (indirect effect a$\beta$ = −0.16, 95% CI [−0.31, −0.02]).

## Conclusions

These results provide novel insights into the drivers of decreased use of protective behaviors over the course of the pandemic, particularly after an asymptomatic or mild COVID-19 infection. More research is needed on the effect of COVID-19 infection on long-term adherence to preventive measures against future pandemics.

---

## Introduction

During the COVID-19 pandemic, potentially protective behaviors such as social distancing, hand washing, and mask wearing were encouraged by public health officials to limit the spread of the virus. Adoption of these behaviors varied along political and ideological lines: research has shown that their use is associated with political party affiliation, COVID-19 conspiracy beliefs, trait reactance, and more [1,2,3]. These behaviors have also been linked to race and ethnicity: White adults are less likely to report mask wearing than racial and ethnic minorities in the United States [4]. Adherence to protective behaviors against COVID-19 across the population has waned over time, a phenomenon known as pandemic fatigue [5]. The drivers of pandemic fatigue are not well understood. Notably, pandemic fatigue increased sharply after the initiation of the vaccination campaign against COVID-19 [6], leading to reductions in COVID-19 mitigation strategies [7]. Little is known, however, about the effect of COVID-19 infection – and the perceived increase in natural immunity after infection– on future use of protective behaviors.

The impact of protective behaviors on risk of COVID-19 infection is well documented in the literature. However, the inverse of this relationship – the impact of personal COVID-19 history on long-term protective behaviors post-infection – remains unclear. People tend to increase their protective behaviors when their perceived risk of infection is high [8,9]. Studies find that individuals wear face masks and/or gloves more often after being exposed to or infected by COVID-19 [10,11] and protective behavior increases as the number of cases in one's family or social network rises [12]. However, a cohort study in Germany did not find the same relationship [13]. The variance in findings in post-infection response may be linked to individual experience with COVID-19 infection severity – Galasso et al. find that experiencing symptoms of COVID-19 infection, or knowledge of symptoms experienced by others, increased adherence to health recommendations [14]. Experience with a more severe COVID-19 infection may increase the perceived threat of the virus and thereby result in greater uptake of protective behaviors.

A potential factor modulating variation in COVID-19 protective behaviors is outlined by protection motivation theory (PMT), which posits that one's appraisal about the probability of a threat, and the efficacy of a coping mechanism, motivate one's decision to adopt protective behaviors [15]. Essentially, an increased perceived severity or vulnerability to the SARS-CoV-2 virus may be associated with better adherence to protective behaviors. This relationship has been demonstrated in the literature [16,17], which finds a consistent link between higher risk perception, perceived severity, and perceived susceptibility to COVID-19 with increased use of protective behaviors [18,19,20]. Applying PMT to pandemic fatigue suggests that vaccination could lessen the perceived threat of the virus due to acquired immunity, thereby reducing protective behaviors. Because nearly 90% of individuals 16 and older in the U.S. are estimated to have infection-induced seroprevalence [21], it is important to understand whether and to what degree past infection status affects protective behavior use, and if that relationship is mediated by changes in the perceived susceptibility of the virus.

This study examines whether awareness of one's COVID-19 infection is associated with change in one's protective behaviors. First, we assessed change in protective behavior use – social distancing, hand washing, and mask wearing – from baseline to follow-up. Then, we tested the relationship between self-reported COVID-19 infection status and use of protective behaviors at follow-up after the infection. We examined the differential effects of COVID-19 symptom severity experienced by those who reported COVID-19 infection on protective behavior use at follow-up. We predicted that individuals who had a mild infection would reduce protective behaviors due to their acquired natural immunity and lower perceived susceptibility to the virus. Conversely, we also tested the competing hypothesis that COVID-19 infection would increase individuals' perceived susceptibility to the virus, which would increase their protective behaviors. To assess the effect of perceived susceptibility as a potential mediating factor, we conducted a secondary analysis focused on whether perceived susceptibility to COVID-19 mediated the relationship between test positivity and protective behaviors.

Our study aims to better understand the drivers of protective behaviors after recovery from COVID-19 infection. As COVID-19 has become endemic, our research will provide key insights into protective behavior use and guide public health strategies for both COVID-19 and future communicable disease outbreaks.

## Methods

### Study procedures

The Los Angeles Pandemic Surveillance Cohort Study researched the experience of COVID-19 among residents in Los Angeles County [22]. A representative sample of 1335 adults were recruited between March 22 and April 23, 2021. The recruitment was conducted through an online/telephone survey by the market research firm, LRW, A Material Company. The current study focused on participants who completed the baseline questionnaire in May 2021 and the follow-up questionnaire in January 2023. Eligible participants were age 18 and above and reported a Los Angeles County residential ZIP code at the time of each survey. All participants provided written informed consent. The survey was available in English and Spanish. The Los Angeles County Department of Public Health institutional review board approved all the study procedures. Study method and results are reported following the Strengthening the Reporting of Observational Studies in Epidemiology (STROBE) guideline [23]. All procedures performed in studies involving human participants were in accordance with the ethical standards of the institutional and/or national research committee and with the 1964 Helsinki declaration and its later amendments or comparable ethical standards.

### Study outcome measures

We measured three protective behaviors both at baseline and at follow-up. For social distancing, we asked "How often are you trying to keep at least 6 feet between you and other people you don't live with to avoid spreading illness?". For hand washing, we asked "After being outside your home or car, how often do you wash or sanitize your hands?". For mask wearing, we asked "When you leave your home, how often do you wear a facemask?" The responses were "never",

"rarely", "sometimes", "often", and "always". For the analysis, we combined "never", "rarely", "sometimes" as a single category to generate a three-level outcome for each behavior, as "never" and "rarely" had very few responses.

For perceived susceptibility to COVID-19, we used the 6-item validated COVID Stress Scale [24]. An example item asked, "in the past 7 days, how worried were you about catching the virus?". The responses were "not at all worried" (0), "slightly worried" (1), "moderately worried" (2), "very worried" (3), and "extremely worried" (4). Cronbach's alpha for baseline items was α = 0.89 and for follow-up was α = 0.91, both with good to excellent internal consistency. We averaged the 6 items in the scale to create a perceived susceptibility score.

### Predictor and covariates

The key predictor was testing positive for COVID-19. Participants who reported ever testing positive in the follow-up questionnaire were also asked about the severity of symptoms they had experienced. We measured levels of symptom severity using the 7-point Global Overall Symptom validated scale [25]. We recategorized responses from the scale and combined that with test positivity to create a three-level predictor: 1) never tested positive for COVID-19, 2) tested positive with asymptomatic or mild symptoms, and 3) tested positive with moderate to very severe symptoms.

Participants reported their age, gender, race/ethnicity, political affiliation, and educational attainment. Participants reported their residential ZIP code at baseline. We adopted the approach in Lam et al. [22] to first convert the ZIP code into Service Planning Area (SPA) of residence in Los Angeles County, then into three multi-SPA COVID-19 Mortality Impacted Areas (CMIA) based on the age-adjusted monthly COVID-19 mortality rates between March 1, 2020 and April 15, 2021 (see Lam et al [22] for the mortality rates). Participants also reported the number of COVID-19 vaccines received, comparing those who were not vaccinated to those without and with booster shots. These factors were included as covariates as they have been linked with protective behaviors against COVID-19 in the literature [2,18,26].

### Statistical analysis

We provided descriptive statistics on participant characteristics and their experience of COVID-19. We then compared the levels of protective behavior use at baseline and follow-up using Wilcoxon signed-rank tests and conducted multivariable ordinal logit regression model for each behavior. The models tested the association between COVID-19 test positivity and (1) use of protective behaviors, and (2) perceived susceptibility to COVID-19. All study covariates and baseline value of the outcome were controlled for in each model. In addition, we conducted a subgroup analysis by CMIA on the associations. All analysis was performed using Stata 15 with α set at 0.05.

Finally, we conducted mediation analysis to examine whether perceived susceptibility partially explained the relationship between test positivity and protective behavior. We calculated the direct and indirect effects for each outcome, and reported results based on the bootstrap bias-corrected percentile confidence interval to determine significance.

## Results

### Participant characteristics and COVID-19 positivity

A total of 724 participants in the cohort study completed both the baseline and follow-up questionnaires. The analytic sample included 676 participants after removing those who reported testing positive for COVID-19 prior to the baseline questionnaire. The analytic sample included 60% female, 45% aged 50 and up, 28% Hispanic and 45% Non-Hispanic White, 74% college or postgraduate degree holders, and 66% Democrats. A third (32%) lived in areas that were highly impacted by COVID-19 mortality. Nearly all (96%) of the study sample reported receiving at least one COVID-19 vaccine and 60% had two or more booster vaccines. The mean perceived susceptibility to COVID-19 score was 1.05 ± 0.89. For COVID-19 infection, 47% reported never testing positive, while 34% had tested positive and experienced moderate to very severe symptoms. (Table 1) There are no missing data for the demographic characteristics. The self-reported dates of COVID-19 positivity peaked during the winter of 2021/2022, summer of 2022 and winter of 2021/2022 (Fig 1).

**Table 1. Participant demographics and COVID-19 characteristics (N = 676).**

|  | n | % |
|---|---|---|
| **Gender** | | |
| Male | 259 | 38% |
| Female | 407 | 60% |
| Non-binary | 10 | 2% |
| **Age Group** | | |
| 18-29 | 50 | 7% |
| 30-49 | 323 | 48% |
| 50-64 | 205 | 30% |
| ≥65 | 98 | 15% |
| **Race/Ethnicity** | | |
| Hispanic | 188 | 28% |
| Non-Hispanic White | 302 | 45% |
| Non-Hispanic Black | 61 | 9% |
| Non-Hispanic Asian | 97 | 14% |
| Non-Hispanic Other | 28 | 4% |
| **Political Affiliation** | | |
| Democrat | 445 | 66% |
| Republican | 58 | 8% |
| Independent | 153 | 23% |
| Other | 17 | 3% |
| **Education** | | |
| High school | 24 | 4% |
| Some college | 148 | 22% |
| College graduate | 298 | 44% |
| Postgraduate | 206 | 30% |
| **COVID-19 Mortality Impacted Areas** | | |
| Low | 165 | 25% |
| Middle | 293 | 43% |
| High | 218 | 32% |
| **Perceived Susceptibility to COVID-19** | | |
| mean ± SD (score range 0–4) | 1.05 ± 0.89 | |
| **COVID-19 Infection and Symptoms** | | |
| Never tested positive for COVID-19 | 319 | 47% |
| Had COVID-19, asymptomatic or mild symptoms | 125 | 19% |
| Had COVID-19, moderate to very severe symptoms | 232 | 34% |
| **COVID-19 Vaccination Status** | | |
| Not vaccinated | 29 | 4% |
| Vaccinated, no booster | 62 | 9% |
| Vaccinated, 1 booster | 180 | 27% |
| Vaccinated, ≥ 2 boosters | 405 | 60% |

## Change in protective behaviors between baseline and follow-up surveys

Adherence to all three forms of protective behavior declined from baseline to follow-up (Table 2). At baseline, 50% of participants reported "always" social distancing, and at follow-up, 13% reported "always" social distancing (p < 0.001).

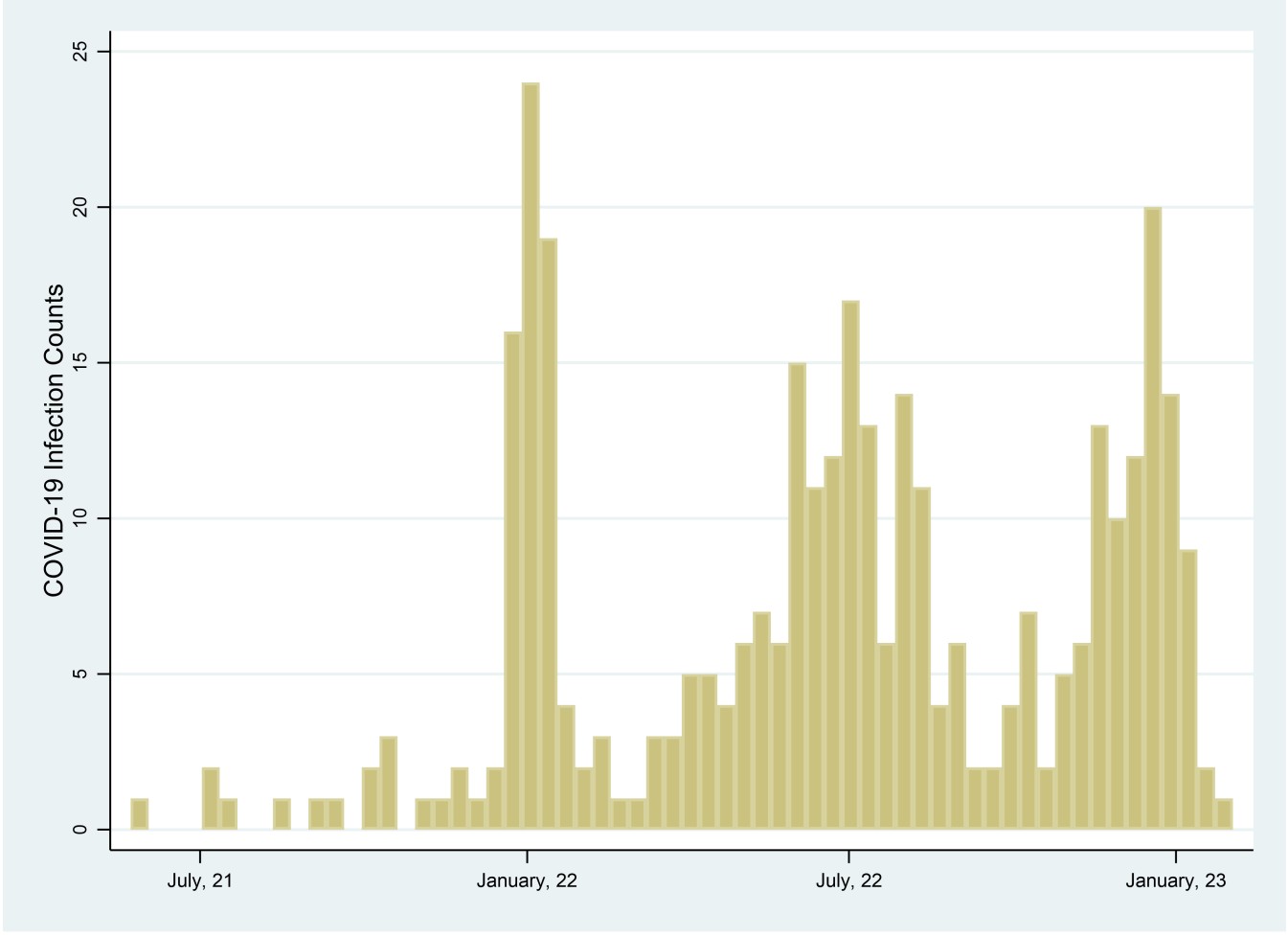

**Fig 1. Timeline of self-reported COVID-19 infection between baseline (May 2021) and follow-up (January 2023) survey.**

**Table 2. Protective behaviors at baseline and follow-up (N = 676).**

|  | Baseline | Follow-up | P value |
|---|---|---|---|
| **Social Distancing** |  |  |  |
| Never/ rarely/ sometimes | 12% | 57% | <0.001 |
| Often | 38% | 30% |  |
| Always | 50% | 13% |  |
| **Hand Washing** |  |  |  |
| Never/ rarely/ sometimes | 11% | 24% | <0.001 |
| Often | 35% | 39% |  |
| Always | 54% | 37% |  |
| **Mask Wearing** |  |  |  |
| Never/ rarely/ sometimes | 4% | 59% | <0.001 |
| Often | 20% | 22% |  |
| Always | 76% | 19% |  |

Baseline: May 2021; Follow-up: January 2023. Wilcoxon signed-rank test performed.

For hand washing, 54% reported "always" hand washing at baseline and 37% at follow up (p < 0.001), and 76% reported "always" mask wearing at baseline, which decreased to 19% "always" mask wearing at follow up (p < 0.001).

### Association between COVID-19 infection and protective behaviors at follow-up

Table 3 displays results of the multivariable ordinal logit regression models examining prior COVID-19 infection and future change in protective behaviors. Compared to those who never tested positive for COVID-19, an asymptomatic or mild COVID-19 infection was associated with less social distancing (aOR = 0.57, 95% CI [0.35, 0.92]) and less mask wearing (aOR = 0.63, 95% CI [0.40, 0.99]) at follow-up. The effect of asymptomatic or mild infection on hand washing was not significant. Moderate to severe COVID-19 infection was associated with less mask wearing (aOR = 0.55, 95% CI [0.38, 0.81]). There was no significant relationship between moderate to severe infection and social distancing or hand washing.

### Association between COVID-19 infection and perceived susceptibility to COVID-19

Table 3 shows that an asymptomatic or mild COVID-19 infection was associated with lower perceived susceptibility to COVID-19 (aβ = −0.17, 95% CI [−0.33, −0.02]). There was no association between moderate to severe infection and perceived susceptibility to COVID-19.

### Association between demographic characteristics and protective behaviors at follow-up

Regression models also identified demographic characteristics associated with protective behaviors at follow-up. Male gender was associated with less hand washing (aOR = 0.48, 95% CI [0.35, 0.66]), less mask wearing (aOR = 0.56, 95% CI [0.40, 0.79]), and lower perceived susceptibility (aβ = −0.18, 95% CI [−0.30, −0.06]) compared to female gender. Age greater than or equal to 65 was associated with more mask wearing (aOR= 2.31, 95% CI [1.04, 5.13]) compared to age 18–29. Non-Hispanic Black adults reported more social distancing (aOR = 1.93, 95% CI [1.07, 3.49]), more hand washing (aOR = 2.70, 95% CI [1.49, 4.80]), and more mask wearing (aOR = 1.93, 95% CI [1.06, 3.50] compared to Non-Hispanic White adults. Having a postgraduate degree was associated with more mask wearing (aOR = 3.53, 95% CI [1.09, 11.5]) compared to high school graduates at follow-up. (Table 3) Participants who lived in middle CMIA were more likely to practice social distancing (aOR = 1.89, 95% CI [1.24, 2.88]) and mask wearing (aOR = 1.60, 95% CI [1.05, 2.44]) compared to those who lived in low CMIA (Table 3).

### Testing perceived susceptibility to COVID-19 as a partial mediator between COVID-19 infection and protective behaviors

Mediation analysis is displayed in Fig 2. The purpose of this analysis was to test whether perceived susceptibility partially explained the relationship between past infection and future use of protective behaviors. Our analysis finds a statistically significant indirect effect of perceived susceptibility between asymptomatic or mild COVID-19 infection and mask wearing, but not for moderate to severe COVID-19 infection. The total direct effect of COVID-19 infection on mask wearing was aβ = −0.84 (95% CI [−1.56, −0.12]). The indirect effect of asymptomatic or mild infection through perceived susceptibility on mask wearing was β = −0.16 (95% CI [−0.31, −0.02]). In sum, perceived susceptibility to COVID-19 partially mediated 15.4% of the effect of asymptomatic or mild COVID-19 on mask wearing. Perceived susceptibility to COVID-19 was not a significant mediator for hand washing or social distancing,

### Subgroup analysis of mask wearing and COVID-19 positivity by COVID-19 mortality impacted areas

Fig 3 shows the unadjusted distribution of mask wearing frequency category at follow-up by COVID-19 test positivity in each CMIA. Overall, participants who had never tested positive for COVID-19 reported more frequent mask wearing than those who had tested positive. However, the subgroup analysis by CMIA revealed that more frequent mask wearing

**Table 3. Association between COVID-19 infection, protective behaviors and perceived susceptibility to COVID-19.**

| | Social Distancing* | | Hand Washing* | | Mask Wearing* | | COVID-19 Susceptibility** | |
|---|---|---|---|---|---|---|---|---|
| | aOR | 95% CI | aOR | 95% CI | aOR | 95% CI | aBeta | 95% CI |
| **COVID-19 Infection and Symptoms** | | | | | | | | |
| Never tested positive for COVID-19 | Ref | | NS | | Ref | | Ref | |
| Had COVID-19, no or mild symptoms | 0.57 | 0.35, 0.92 | | | 0.63 | 0.40, 0.99 | -0.17 | -0.33, -0.02 |
| Had COVID-19, moderate to very severe symptoms | 0.77 | 0.53, 1.12 | | | 0.55 | 0.38, 0.81 | -0.04 | -0.18, 0.09 |
| **COVID-19 Vaccination Status** | | | | | | | | |
| Not vaccinated | Ref | | NS | | NS | | NS | |
| Vaccinated, no booster | 0.36 | 0.13, 0.99 | | | | | | |
| Vaccinated, 1 booster | 0.52 | 0.22, 1.23 | | | | | | |
| Vaccinated, ≥2 boosters | 0.50 | 0.21, 1.16 | | | | | | |
| **COVID-19 Mortality Impacted Area** | | | | | | | | |
| Low | Ref | | NS | | | Ref | NS | |
| Middle | 1.89 | 1.24, 2.88 | | | 1.60 | 1.05, 2.44 | | |
| High | 1.55 | 0.98, 2.43 | | | 1.51 | 0.96, 2.37 | | |
| **Gender** | | | | | | | | |
| Female | NS | | Ref | | Ref | | Ref | |
| Male | | | 0.48 | 0.35, 0.66 | 0.55 | 0.39, 0.78 | -0.18 | -0.30, -0.06 |
| Non-binary | | | 1.18 | 0.35, 3.98 | 0.92 | 0.23, 3.61 | -0.16 | -0.62, 0.31 |
| **Age Group** | | | | | | | | |
| 18-29 | NS | | NS | | Ref | | NS | |
| 30-49 | | | | | 1.20 | 0.60, 2.40 | | |
| 50-64 | | | | | 1.82 | 0.87, 3.74 | | |
| ≥65 | | | | | 2.39 | 1.07, 5.34 | | |
| **Race/Ethnicity** | | | | | | | | |
| Hispanic | 1.28 | 0.86, 1.93 | 1.60 | 1.10, 2.33 | 1.32 | 0.87, 1.99 | NS | |
| Non-Hispanic White | Ref | | Ref | | Ref | | | |
| Non-Hispanic Black | 2.09 | 1.14, 3.81 | 2.64 | 1.47, 4.77 | 2.04 | 1.12, 3.72 | | |
| Non-Hispanic Asian | 0.86 | 0.52, 1.43 | 1.07 | 0.68, 1.70 | 1.46 | 0.89, 2.40 | | |
| Non-Hispanic Other | 1.46 | 0.65, 3.27 | 1.25 | 0.58, 2.70 | 3.04 | 1.44, 6.40 | | |
| **Political Affiliation** | | | | | | | | |
| Democrat | NS | | NS | | NS | | NS | |
| Republican | | | | | | | | |
| Independent | | | | | | | | |
| Other | | | | | | | | |
| **Education** | | | | | | | | |
| High school | NS | | NS | | Ref | | NS | |
| Some college | | | | | 2.60 | 0.79, 8.57 | | |
| College graduate | | | | | 3.00 | 0.93, 9.65 | | |
| Postgraduate | | | | | 3.53 | 1.09, 11.5 | | |

*Multivariable ordinal logit regression model, **Multivariable linear regression model.

All models controlled for the same protective behavior or COVID-19 susceptibility measured at baseline.

NS: not significant; aOR: adjusted odds ratio; aBeta: adjusted Beta.

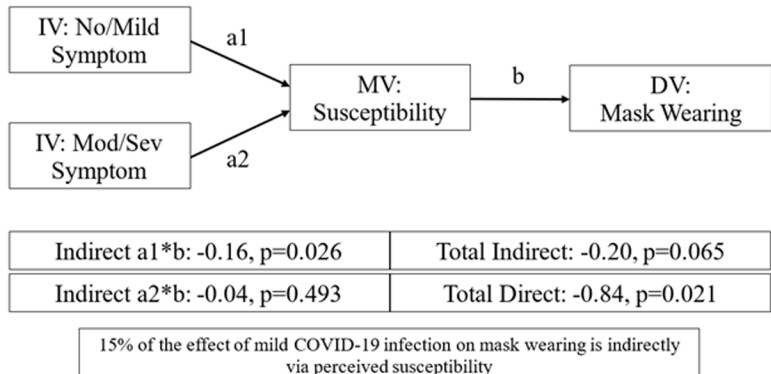

| Indirect a1*b: -0.16, p=0.026 | Total Indirect: -0.20, p=0.065 |
| Indirect a2*b: -0.04, p=0.493 | Total Direct: -0.84, p=0.021 |

15% of the effect of mild COVID-19 infection on mask wearing is indirectly via perceived susceptibility

**Fig 2. Testing perceived susceptibility to COVID-19 as a partial mediator between COVID-19 infection and mask wearing.** IV: independent variable, MV: mediation variable, DV: dependent variable, Mod/Sev: moderate/severe.

among the never tested positive was driven mainly by those who lived in high CMIA. This divergence in behavior is not observed for social distancing and hand washing.

## Discussion

Our study observed a drastic reduction in three potentially protective behaviors against COVID-19 – social distancing, hand washing and mask wearing – from May 2021 to January 2023. Maintenance of protective behavior at follow-up was associated with demographic factors, including non-Hispanic Black ethnicity, a postgraduate degree, female gender, and living in an area more affected by COVID-19 mortality. Notably, reported COVID-19 infection was associated with reduction of future protective behavior use. Individuals who experienced asymptomatic or mild COVID-19 reported less maintenance of all three behaviors after infection, and those who reported moderate to severe symptoms reported less mask wearing. Those who reported asymptomatic or mild COVID-19 reported a lower perceived susceptibility to COVID-19 after infection. Results from the mediation analysis showed that the relationship between mild COVID-19 infection and less mask wearing could be partially explained by a decrease in perceived susceptibility to the virus.

The reduction in COVID-19 protective behaviors throughout the pandemic can be explained by pandemic fatigue – a phenomenon described by the WHO as "demotivation to follow recommended protective behaviors" [6]. However, the association between COVID-19 infection (and perceived natural immunity post-infection) and change in these behaviors is less clear. We found that, in line with our hypothesis, asymptomatic or mild infection led to a reduction in perceived susceptibility and the use of two of three protective behaviors. However, we did not find an increase in perceived susceptibility and in protective behavior after moderate to severe infection. In fact, we observed a decrease in one measure of protective behavior (mask wearing) after moderate to severe infection. A potential explanation for the divergence in results is the time to follow-up. As presented in O'Connell et al., individuals increase their use of protective behaviors within the first week after testing positive [11]. It is possible that a week after infection, the experience of COVID-19 is more immediate, and therefore perceived susceptibility to the virus is higher. Weeks or months after infection the perceived susceptibility effect from the initial infection, even if moderate or severe, may have faded. It is also possible that surviving a severe illness such as COVID-19 may also grant individuals a greater belief in their acquired immunity, especially as the immediate experience of symptoms fades. The other major motivator for protective behavior use is protecting others from infection, which likely decreases with more time from acute infection.

Our results on experience of COVID-19 symptom severity and perceived susceptibility support our hypothesis and PMT. Individuals who have not experienced COVID-19 infection have their perception of the virus shaped by people

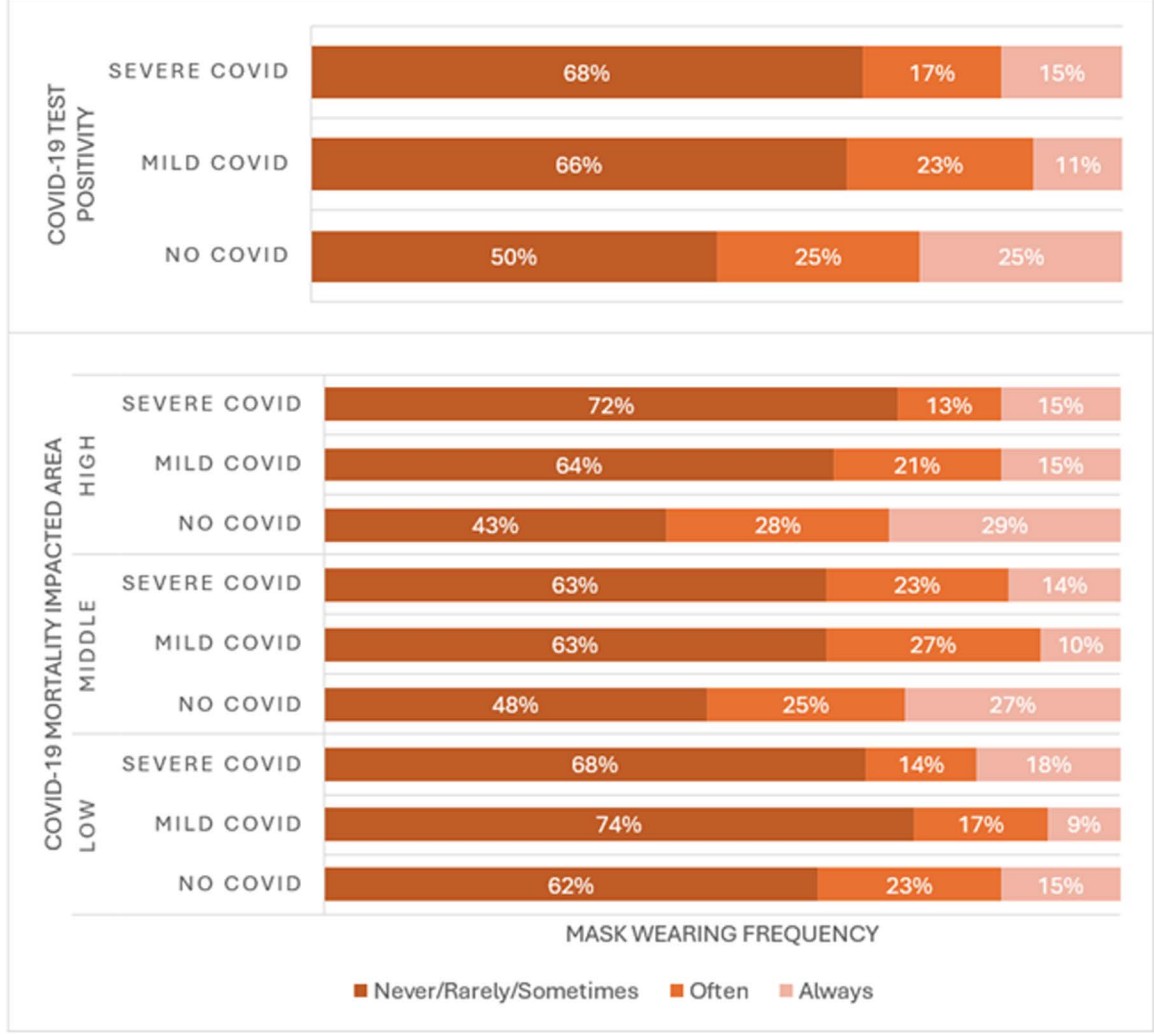

**Fig 3. Subgroup analysis of mask wearing and COVID-19 test positivity by COVID-19 mortality impacted areas.**

around them and by the media, which often focus on the severity of the pandemic. Cipoletta et al. found that COVID-19 cases in one's community and among close contacts increase one's perceived threat of the virus [27]. However, individual experience with asymptomatic or mild COVID-19 infection may have the opposite effect on perceived susceptibility. An individual may conclude, for example, that protective behaviors were unnecessary as they contracted COVID-19 regardless and the infection course was relatively mild. The reduction in one's protective behavior (i.e., mask wearing) after mild infection could also be driven by a reduction in perceived susceptibility to the virus. This reduction is likely due to multiple factors, including the perception of increased natural immunity after a mild infection, and the perception of COVID-19 as less dangerous after experiencing few or no symptoms. Those unaware of their infection status would continue to base their judgement and risk perceptions from other's experience of COVID-19 [14], and as we observed, are more likely to continue adhering to protective behaviors as preventive measures. In addition, participants who lived in areas that were worse hit by COVID-19 mortality continued to practice social distancing and mask wearing, especially those who never tested positive for COVID-19. This result aligns with PMT, as living environments and awareness of risk can motivate the adoption of protective behaviors.

A possible future area of study is understanding other drivers of lessened protective behavior use after asymptomatic or mild infection, as our analysis finds that only about 15% of the effect is due to reductions in perceived susceptibility. It is possible that there are components of pandemic fatigue not directly related to changes in perceived severity of the virus, including changes in beliefs about the efficacy of protective behaviors. It is also possible that we did not capture the total effect of perceived risk, which includes perceived severity, as changes in these perceptions may be subconscious and not identified and reported by participants. Studies that integrate the psychological mechanisms behind perceived susceptibility and pandemic fatigue may provide context to these findings. More research is required on the effect of COVID-19 infection on adherence to other forms of preventive measures, including stay-at-home orders.

While most of the demographic differences in protective behaviors we observed are in accordance with the literature – post-graduate degree attainment, Black race, and female gender are all linked to more protective behavior use [4,10,11] – we did not replicate a well-studied relationship between party affiliation and protective behavior [28,29]. This may be due to our time to follow up, as accumulating pandemic fatigue over the course of the pandemic may have lessened differences in protective behavior use by political parties. It is also possible that our study lacked statistical power to capture this relationship, as only 58 participants (9%) reported a Republican party affiliation. In line with findings on party affiliation, on an individual level, belief in misinformation has been shown to negatively predict adherence to protective behavior and vaccination, adding support for perceived susceptibility as a major driver of protective behavior [30].

Our findings contain key implications for public health responses to future pandemics. Differences in protective behavior adherence by demographic characteristics (e.g., lower adherence in men and younger people) underscore the need for tailored messaging to specific demographic groups to promote adherence. Funding also could be allocated to higher-risk areas and populations. Messaging could target perceived susceptibility as a driver of lower protective behavior use; for example, highlighting the long-term risks of infection may heighten perceived susceptibility and sustain protective behavior use after infection. Initiatives like the WHO's Preparedness and Resilience for Emerging Threats (PRET), which aims to build international collaboration to prepare for future pandemics, may consider integrating specific regional data on protective behavior adherence to better tailor interventions [31].

This study has several limitations. First, time to follow-up since infection is not captured in the statistical model. In theory, individuals who have longer elapsed time since infection may report a greater reduction in protective behaviors. Our goal is to present a parsimonious model that explains the association between COVID-19 infection and adherence to behavior, while accounting for participant baseline level. Second, we did not account for the effect of multiple infections on pandemic fatigue. One study, for instance, shows higher levels of perceived susceptibility to COVID-19 after multiple COVID-19 infections [32], and another study finds that pandemic fatigue is positively associated with personal fear of COVID-19 [33]. Thus, individuals with multiple COVID-19 infections may sustain protective behaviors for an extended period. Third, our sample is specific to Los Angeles County, which may limit external validity to other geographic locations with different socioeconomic status and racial and ethnic composition.

This study is one of the early investigations to examine the effect of COVID-19 infection, symptom severity, and perceived susceptibility, on long-term adherence to protective behaviors. Our findings provide insights into the drivers of demotivation to follow recommended behaviors and may aid public health efforts around COVID-19 and future pandemics. As nations continue to experience surges in COVID-19 with more infectious strains, the possibility of reinfection and severe sequalae, including heightened risk for mental illness, COVID-19 remains a key public health concern and warrants continued research efforts [34].

## Author contributions

**Conceptualization:** Chun Nok Lam, Jennifer B. Unger, Neeraj Sood.

**Formal analysis:** Chun Nok Lam.

**Project administration:** Chun Nok Lam, Jennifer B. Unger, Neeraj Sood.

**Supervision:** Jennifer B. Unger, Neeraj Sood.

**Visualization:** Chun Nok Lam.

**Writing – original draft:** Chun Nok Lam, Shirin Emma Herzig.

**Writing – review & editing:** Chun Nok Lam, Nikhilesh Kumar, Jennifer B. Unger, Neeraj Sood.

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
