## [Decision Letter · Decision Letter 0]

Dear Dr. Lam,

Thank you for submitting your manuscript to PLOS ONE. After careful consideration, we feel that it has merit but does not fully meet PLOS ONE’s publication criteria as it currently stands. Therefore, we invite you to submit a revised version of the manuscript that addresses the points raised during the review process.

Your manuscript was reviewed by two experts in the field. Both identified some problems in your submission which require your attention. Please review the attached comments and provide point-by-point responses. 

We look forward to receiving your revised manuscript.

Kind regards,

Yury E Khudyakov, PhD

Academic Editor

PLOS ONE

Journal Requirements:

https://jou2. rnals.plos.org/plosone/s/file?id=ba62/PLOSOne_formatting_sample_title_authors_affiliations.pdf

5. In this instance it seems there may be acceptable restrictions in place that prevent the public sharing of your minimal data. However, in line with our goal of ensuring long-term data availability to all interested researchers, PLOS’ Data Policy states that authors cannot be the sole named individuals responsible for ensuring data access (http://journals.plos.org/plosone/s/data-availability#loc-acceptable-data-sharing-methods ).

Reviewers' comments:

Reviewer's Responses to Questions

**Comments to the Author**

1. Is the manuscript technically sound, and do the data support the conclusions?

Reviewer #1: Yes

Reviewer #2: Yes

2. Has the statistical analysis been performed appropriately and rigorously?

Reviewer #1: Yes

Reviewer #2: I Don't Know

3. Have the authors made all data underlying the findings in their manuscript fully available?

Reviewer #1: Yes

Reviewer #2: No

4. Is the manuscript presented in an intelligible fashion and written in standard English?

Reviewer #1: Yes

Reviewer #2: Yes

Reviewer #1: The study present a very important aspect of pandemic management. Please:

1- Reflect on the rational of the study.

2- Detail the impact of this study on macro and micro levels

3- Link findings of this study with the ongoing WHO activities to prepare for fighting pandemic X

4- Expand the discussion section.

Reviewer #2: Interesting paper

The participants are almost all vaccinated. it would be interesting to see what the pattern is in unvaccinated participants.

Similarly, subgroup analysis by territory/areas worst hit by COVID vs those less affected will make the paper better

**Do you want your identity to be public for this peer review?** For information about this choice, including consent withdrawal, please see our Privacy Policy

Reviewer #1: **Yes: ** WEAM BANJAR

Reviewer #2: No

---

## [Author Response · Author response to Decision Letter 1]

15 May 2025

Reviewer #1:

Reflect on the rational of the study.

Response: Thank you for this suggestion. We have added a sentence in the introduction to be exact about the exact rationale and purpose for our study:

“Our study aims to better understand the drivers of long-term protective behavior use after people recover from COVID-19 infection. As COVID-19 has become endemic, our research will provide key insights into protective behavior use and guide public health strategies for both COVID-19 and future communicable disease outbreaks.”

We have also added context to the conclusion summarizing why we conducted this study, which is outlined in our response to the second reviewer’s suggestion.

Detail the impact of this study on macro and micro levels.

Response: Thank you for pointing out an area where we can add valuable context to our study.

We detail the impact of the study on macro levels by outlining how our study can inform policy to curb future pandemics. Please see:

“Our findings contain key implications for public health responses to future pandemics. Differences in protective behavior adherence by demographic characteristics (e.g., lower adherence in men and younger people) underscores the need for tailored messaging to specific demographic groups to promote adherence. Funding may also be allocated to higher-risk areas and populations. Messaging may also need to target perceived susceptibility as a driver of lower protective behavior use – highlighting the long-term risks of infection, for example – may heighten perceived susceptibility and sustain protective behavior use after infection. Initiatives like the WHO’s Preparedness and Resilience for Emerging Threats (PRET), which aims to build international collaboration to prepare for future pandemics, may consider integrating specific regional data on protective behavior adherence to better tailor interventions.”

The impacts of our study on micro levels have also been added throughout. For one, we added analysis throughout the paper examining the impacts of living in high-mortality areas with adherence to protective behaviors, showing how individual behavior changes based on environment. Additionally, we add reference to studies linking individual beliefs to protective behavior use, including a study that found that belief in misinformation around the pandemic was a negative predictor of protective behavior use and vaccination. We explain in the discussion how our results provide insight into the way identity and personal characteristics can be linked to protective behavior use and can inform how interventions to increase protective behavior are designed and targeted.

Link findings of this study with the ongoing WHO activities to prepare for fighting pandemic X

Response: As outlined in our response to the last suggestion, we link our discussion of future policy to the WHO Preparedness and Resilience for Emerging Threads (PRET) program which is in use to prepare for future pandemics.

Expand the discussion section.

Response: Please see the prior responses to see how we have expanded the discussion.

Reviewer #2:

The participants are almost all vaccinated. it would be interesting to see what the pattern is in unvaccinated participants.

Response: Results from bivariate analyses show that participants who were not vaccinated were more likely to report always practicing physical distancing (21%) than other vaccination groups (13%). They were also more likely to always wear masks (24%) compared the 1 vaccine group and the 1 booster group (10%). However, when controlling for other covariates, especially the baseline protective behavior level, the difference on protective behaviors at follow-up by vaccination status was no longer significant (Table 3).

Similarly, subgroup analysis by territory/areas worst hit by COVID vs those less affected will make the paper better.

Response: Thank you for the great suggestion. We adopted the approach used in Lam et al. (2024) to include regional assessment on COVID-19’s impact on the community based on death rates during the first year of the pandemic. We included the newly added variable, COVID-19 mortality impacted area, as a covariate in the regression models, as well as in a subgroup analysis analyzing the association between mask wearing and COVID-19 test positivity. Multiple changes have been made to the methods, results and discussion sections. We found that participants who lived in areas that were hit hardest by COVID-19 mortality were more likely to practice social distancing and wear masks. In the subgroup analysis, now shown in Figure 3, the more frequent use of masks in the hardest hit area was among those who had never tested positive for COVID-19. This finding supports the adoption of protective behavior based on one’s perceived immunity and perceived susceptibility of COVID-19 based on positivity status and surrounding environment.

---

## [Editor Report · Decision Letter 1]

Associations between COVID-19 Infection, Symptom Severity, Perceived Susceptibility, and Long-Term Adherence to Protective Behaviors: The Los Angeles Pandemic Surveillance Cohort Study

PONE-D-24-51022R1

Dear Dr. Lam,

We’re pleased to inform you that your manuscript has been judged scientifically suitable for publication and will be formally accepted for publication once it meets all outstanding technical requirements.

Kind regards,

Yury E Khudyakov, PhD

Academic Editor

PLOS ONE
---

## [Editor Report · Acceptance letter]

PONE-D-24-51022R1

PLOS ONE

Dear Dr. Lam,

I'm pleased to inform you that your manuscript has been deemed suitable for publication in PLOS ONE. Congratulations! Your manuscript is now being handed over to our production team.

Kind regards,

on behalf of

Dr. Yury E Khudyakov

Academic Editor

PLOS ONE